# SAWIT—Security Awareness Improvement Tool in the Workplace

**Ana Kovačević** [1,*] and **Sonja D. Radenković** [2,*]

1 Faculty of Security Studies, University of Belgrade, Gospodara Vučića 50, 11000 Belgrade, Serbia
2 Belgrade Banking Academy—Faculty of Banking, Insurance and Finance, Union University, Zmaj Jovina 12, 11000 Belgrade, Serbia
* Correspondence: ana.kovacevic@fb.bg.ac.rs (A.K.); sonja.radenkovic@bba.edu.rs (S.D.R.)

**Abstract:** Cyberattacks are becoming increasingly sophisticated and severe, and an organization's protection depends on its weakest member. Although users are aware of the risks in cyberspace, most of them do not follow best practices, and there is a need for permanent structured training. The majority of previous training programs concentrated on technically educated users, but the organization is only as secure as the most vulnerable link in it. The paper presents SAWIT, a new Web tool, created with the goal of improving security awareness among employees. It is an innovative artificial intelligence framework aimed at improving the cyber security knowledge of employees by using collaborative learning and assessment within the specified knowledge transformation model.

**Keywords:** cyber security; security awareness; knowledge management; collaborative learning; assessment

## 1. Introduction

Today, it is difficult to imagine life without information technology, but there is also a dark side to it. The number of attacks in cyberspace has grown in the last few years, and they are likely to increase in the future. According to PricewaterhouseCoopers there were 42.8 million security incidents in 2016, which was a rise of 48% compared to the previous year [1]. The Ponemon Institute in 2017 estimates the economic impact of security breaches at nearly half a trillion dollars globally [2]. In addition to the growth of attacks, they are also becoming more and more sophisticated and severe, such as Stuxnet [3].

As we have seen, the positional costs of cyber security attacks are extremely high, so it is necessary to adequately protect organizations. More organizations today have realized the need to invest not only in technical protection but also in employee awareness. The report compiled by the Sans Institute in 2016 stated that larger organizations spend almost 35% of their annual security budget on end user security training and awareness [4]. With the wide-scale adoption of information technologies in recent decades, the profile of the end user has also changed. The average user of information technology is not necessarily technically educated and is unlikely to have studied cyber security in his/her previous education. Because of that, the organization needs to invest in the security awareness of employees in order to inform or control its own systems [5].

Although employees use computers and the Internet regularly and may be somewhat aware of security risks, the majority of them are unsure of the measures required to achieve the confidentiality, integrity and availability of information in cyber space. For example, even if they have heard about phishing, some users are not sure how to recognize the problem or react appropriately. According to research conducted by the Pew Research Center, only 54% of the respondents correctly identified phishing attacks [6]. According to the Ponemon Institute report, careless humans caused 78% of cyber-attacks [2]. Parson et al. stated that the naïve behavior and accidental mistakes of computer users

are the most frequent reasons for cyber security incidents [7]. Unaware employees may share sensitive information with unauthorized persons or inadvertently install malware, create weak passwords or be the victim of the phishing attack. The CIA (Central Intelligence Agency) discovered that 47 government agencies had been compromised, and hackers had gained access to over 21 million government employee accounts [8]. It is necessary to maintain a high level of security awareness among all employees, not only among those whose jobs are IT-related [9]. With the rising level of attacks in cyber space, it is critical for organizations to raise the level of security awareness.

"Cyber security is a computing-based discipline involving technology, people, information, and processes to enable assured operations in the context of adversaries" [10] (p. 16). The need for the existence of cyber security was evident with the first computers. In the past, organizations were primarily focused on investing in technical protection, and uneducated employees thus became the most vulnerable. Nowadays, organizations are vulnerable because of human error [11].

Threats to the cyber world are becoming more sophisticated and they are constantly changing, making it necessary to continuously educate users/employees to adapt to the changes in technology and threats. Humans are the central figures in cyber security awareness, and reducing the risks in cyber space demands raising security awareness among the population. It is very important to raise the level of security culture among employees, and one of the solutions for this is raising cyber security awareness, which will be both flexible and adaptable enough to meet the new requirements in cyber space.

In this paper, we propose a tool for raising cyber security awareness called SAWIT—a tool for improving security awareness in the workplace. The main idea was to enhance cyber security among both end users and IT professionals in order to raise the level of security awareness in organizations. A large amount of money is spent on investing in technology to protect organizational assets. Nevertheless, it is very important to invest in the education of employees, since the company's information security team cannot provide all the necessary security measures for all kinds of threats. In other words, it is necessary to train employees in the area of cyber security in an appropriate way during everyday activities in the workplace.

The paper is organized as follows. In the following section, security awareness is presented as is its importance to society, as well as the main concepts of the security awareness software, while the subsequent section presents previous research. The next section presents some application scenarios of the SAWIT software. The conceptual framework of the SAWIT software is provided in the following section. In the penultimate section, we discuss our work in the context of related work, before concluding the paper in the final section.

## 2. Security Awareness

In this section, we present security awareness and its importance for society and introduce the main concepts of the security awareness software.

### 2.1. Security Awareness

Security awareness is defined in NIST Special Publication 800-16 as follows: "Awareness is not training. The purpose of awareness presentations is simply to focus attention on security. Awareness presentations are intended to allow individuals to recognize IT security concerns and respond accordingly" [12] (p. 15). It is very important to emphasize that security awareness should be a continual process because new attacks appear constantly. Security awareness is more closely defined by using the term situation awareness [13], i.e., "the perception of the elements in the environment within a volume of time and space, the comprehension of their meaning and the projection of their status in the near future".

The leading idea is to achieve of awareness in a given situation, and the model consists of three levels:

- Perception (recognition of the status, attribute and dynamics of related elements in environment);

- Comprehension (synthesis the separated perceptual elements through analysis and evaluation process based on the previously collected data);
- Projection (prediction of analyzed information obtained in comprehension level) [13].

Situational awareness applied to cyber domain is *cyber-situational awareness* and can be defined as "compilation, processing and fusing of network data to understand a network environment and accurately predict and respond to potential threats that might occur" [14].

Although situational awareness was firstly mentioned twenty-five years ago, it is still a hot research topic applied to a broad range of research areas. Recently Park et al., have proposed a framework to measure the risk of Internet of things devices based on situation awareness [15]. On the other hand, security awareness serves to enhance the perspective that new cyber sensors can contribute to situational awareness for the purpose of understanding what needs to be done in terms of the desired effects and the actions that ought to be undertaken to achieve these effects [5].

According to the National Institute of Standards and Technology [16], the first step in information security learning is awareness, the next is training, and the final stage is education. Security awareness offers organizations numerous advantages, such as compliance with the increasing number of laws and regulations regarding forms of training and awareness activities, achieving customer trust and satisfaction, enhancing accountability and compliance, reducing the number of security incidents and improving corporate reputation [17,18].

User factors such as attitude, self-efficacy and the perceived response costs of security tasks are linked to information security policies [19]. On the other hand, a lack of cyber security knowledge could result in users' lack of confidence, higher dissatisfaction and a sense of helplessness, as indicated by Salanova et al. [20]. According to Pawlowski and Jung, users who read about cyber security terms and concepts are more aware about cyber security issues [21].

Ernst & Young analyzed the ability of companies to meet the requirements for building security awareness in 2013, and three years after: Both reports confirmed the lack of skilled resources [22,23]. According to Oracle and KPMG's report from 2019, the shortage of cyber security skills is a major factor contributing to the inability to process security events, and 53% of organizations stated that they have a problematic shortage of skills in the area of cyber security [24]. Thus, one of the solutions for this problem may be technology enhanced learning. Alotaibi and Alfehaid stated that web-based programs raise security awareness in tested domains (password, email and virus management) [18].

### 2.2. Security Awareness Software

Today there are a great number of public options (i.e., websites, booklets, info graphics, blogs, posters, video clips), which are valuable resources for raising the level of security awareness (e.g., https://www.stopthinkconnect.org). Although a great deal of information is available, at different levels, there is no clear standardized approach to this issue, and such information may be very demanding and confusing to newcomers, especially to the non-technical layperson.

Korovessis et al. stated that although all materials proved to be good for security professionals for creating awareness, they are not appropriate for individuals who are new in the field of security and should be protected from potential threats [17]. There is a need for a more structured learning approach, which would combine valuable resources in a more efficient way and build an environment where employees can learn about security concepts in a more active manner and become competent and confident users of technology [17]. Consequently, it is necessary to apply the knowledge creation process to cyber security issues where individuals can create and share knowledge with each other and thus promote the growth of knowledge through a continuous and dynamic process.

The process of knowledge creation and evaluation should be applied by using the Nonaka and Takeuchi model, known as the SECI model (derived from the words Socialization, Externalization, Combination and Internalization) [25]. This is a well-known model for knowledge-building situations. It describes the process of knowledge acquisition from the initial need and motivation, through the transformation of that need into learning activities (Socialization), collaboration with other learners

(Externalization), the application of the newly learned skills in a real-world situation (Combination), as well as gaining more experience with the new skills and thus improving performance (Internalization). The SECI model is the theoretical foundation for understanding the relationship between KM and intellectual capital, and it outlines different interactive spaces, in which tacit knowledge can be made explicit [26].

Another issue that should be considered is the data model that needs to be applied. Following the principles of artificial intelligence, it is very important to create software that conceptualizes a cyber security domain in a formal and explicit manner [27]. Because of that, the software needs to use the cyber security ontology, as "an explicit specification of a conceptualization", mapping the existing knowledge in order to present a holistic view of the cyber security domain [28].

Accordingly, a more structured program for security awareness should be made, which is user-centric, or adjusted to the specific user, his/her previous background, his/her work requirements. Pharm et al. emphasize that knowledge sharing among employees can be effective in improving awareness and security compliance and is important for business success [29]. The same opinion is shared by Safa and Von Solms, who discuss the importance of experts and staff with information security knowledge capabilities as effective in enhancing security behavior [19].

In order to be more effective, cyber security awareness material should be interesting, current and simple; in addition, it should also be targeted, actionable and doable and provide feedback [30]. The user's impression of the suitability of the software is also very important in terms of acceptance and improving cyber security culture.

Security awareness software should fulfill several requirements [17]:

- Have a structured approach.
- Measure existing user knowledge and improve it in future knowledge creation.
- Be structured and modular, so new items should be added.
- Be interesting and interactive, so users are more engaged.
- Be web-based, so it can be accessed from anywhere with minimal information resources.

## 3. The SAWIT Tool Experience

In order to enhance cyber situational awareness [31] while performing work-related activities, an employee runs the SAWIT tool accessible from his/her web browser. The tool runs from within a Chrome web browser, from the so-called SAWIT toolbar (Figure 1).

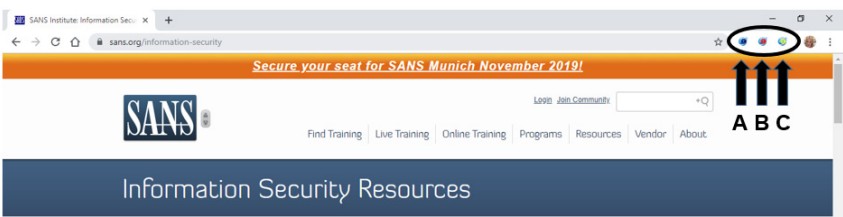

**Figure 1.** SAWIT add-on toolbar. (**A**) Knowledge Sharing; (**B**) Security Resources; (**C**) Assessment

SAWIT button A in the upper right corner (Figure 1A) opens the SAWIT Knowledge Sharing dialog (Figure 2A), filling it with the URL as well as the title of the opened security awareness resource. The initialization process in this action covers the data about the user who performs the knowledge sharing, the knowledge resource itself as well as the web page to be bookmarked/tagged or annotated. If the current security resource has already been bookmarked by another employee, the *Also Tagged by* part (Figure 2A) will show his/her name. By clicking on the user shown in this part, the SAWIT tool will show the current security resources of the specified user.

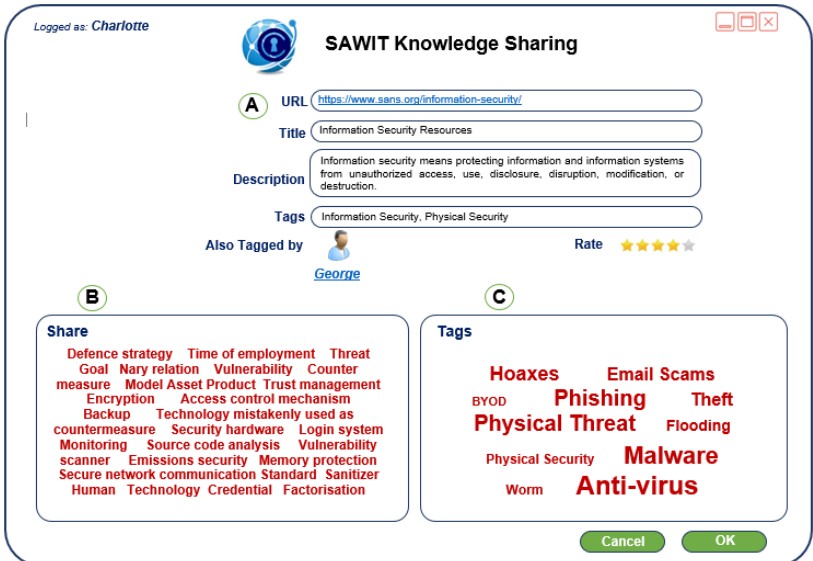

**Figure 2.** SAWIT knowledge sharing dialog. (**A**) Dublin Core Terms (**B**) Security Domain Concepts (**C**) User-defined Tags.

Sharing cyber security resources in the SAWIT tool refers to describing them with metadata (Dublin Core, http://dublincore.org/documents/dcmi-terms/, Terms and the like), concepts from the cyber security ontology (semantic annotation) and tagging. There are three types of annotations that can be performed in the SAWIT tool:

- Annotation by using the concepts of cyber security ontology [32]. By using the artificial intelligence techniques, specifically knowledge representation and inference methods [33], employees get the knowledge resources that are described with the concepts of cyber security domain (security resources). This process is automated, and it is performed by using the KIM semantic annotation platform [34]. The results are shown in the *Share part* (Figure 2B). Semantic annotations given in that way offer two kinds of benefits regarding the current security resource: enhanced information retrieval and improved interoperability.
- The SAWIT tool performs the process of describing a security resource with the Dublin Core Terms vocabulary. This service instantiates the metadata of the corresponding security resource and stores metadata elements in the Title, Author(s), Subject and Description (Figure 2A).
- The third type of annotation is tagging. It is done manually, during the process of bookmarking a security resource (and also when updating it), by adding tags that additionally describe the security resource. User-defined tags stored in the repository are presented in the *Tags part* (Figure 2C).

Clicking button B on the SAWIT add-on in the browser (Figure 1), the employee opens the *SAWIT Security Resources* dialog (Figure 3), which presents the current cyber security materials that can be used by employees in the preparation process for the final tests. At the beginning of using the SAWIT tool, there is the repository security resources given by the organization, which is the starting point for all employees to improve cyber security awareness [31]. By using the SAWIT tool, the repository is being enriched with knowledge assets allowing employees to browse it and view individual assets so as to perform a semantic search of the repository. The search functionality (Figure 3) in the SAWIT tool includes a semantic search and faceted search.

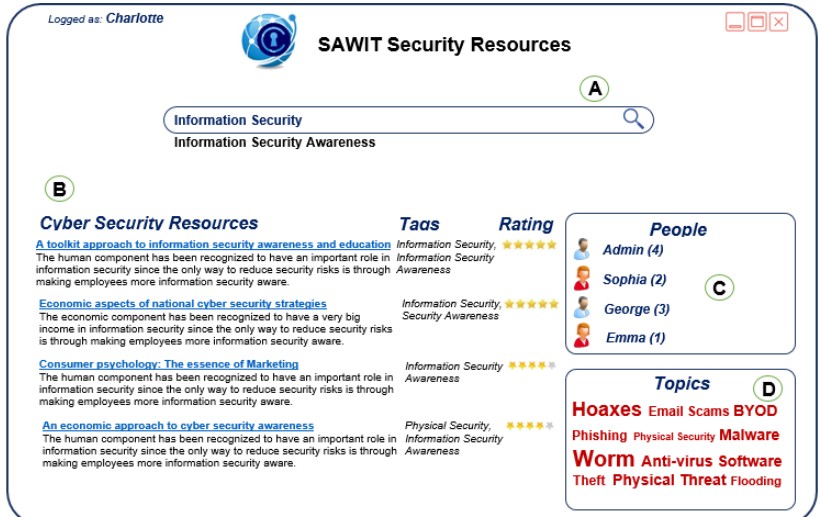

**Figure 3.** SAWIT Security Resources dialog (**A**) Semantic Search Engine (**B**) Cyber Security Resources (**C**) Search by People (**D**) Search by Topics.

The semantic search functionality of the SAWIT tool enables the effective and easy retrieval of stored cyber security assets through an understanding of the employee's profile and the contextual meaning of the search terms. It allows the user to search for specific cyber security resources based on a given input, such as selected domain concepts or tags. The semantic search finds all the related cyber security resources for the given input. If none of the available cyber security resources directly matches the user's request, the semantic search will check for semantically related domain concepts or tags, find cyber security assets annotated with them and suggest those as potentially useful assets. In order to find similar domain concepts, the semantic search uses existing ontology relations, as well as domain concepts and tags frequently used in the same context. All the cyber security assets discovered in this way are further compared with the employee's profile in order to compute the relevance of each asset for the given employee. Thus, the employee is presented with the most relevant cyber security asset in the first position.

Faceted search in the SAWIT tool leverages metadata fields and values to provide employees with visual facets (options) for clarifying and refining query results. In Figure 3, the most important facets for the given context are *People* (Figure 3C) and *Topics* (Figure 3D) given from the most used tags in the repository. They are opened to support users' information-seeking activities, and this kind of support is often more effective than best-first search.

Clicking the C button on the SAWIT add-on in the browser (Figure 1), the employee opens the *SAWIT Assessment* dialog (Figure 4), which enables employees to be assessed by using the final tests, prepared according to the IMS-QTI standard (https://www.imsglobal.org/question/index.html).

In the assessment process, the employees have the opportunity to answer two types of question: single choice and multiple choice. The process of creating the assessment tests is automated, since the SAWIT tool is composed of the assessment item and assessment test tool [35]. The administrator is in charge of creating the assessment items, as well as the testing criteria. As soon as the employee starts the assessment test process, the SAWIT tool leverages all the requirements and creates the assessment test by using the proposed assessment items. Upon finalizing the test, the employee gets the results that he/she is free to accept or ignore. If the employee accepts the results, they are saved in the database under the employee's profile. Otherwise, if the employee rejects them, he/she will perform the testing process again, but the employee's testing attempt is saved in the database.

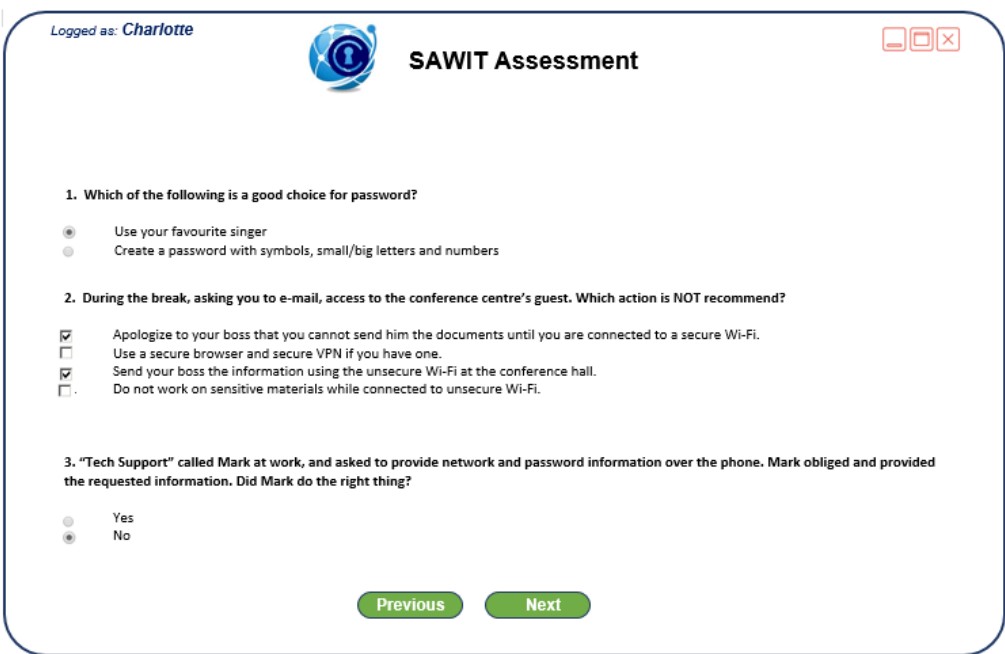

**Figure 4.** SAWIT assessment dialog.

## 4. The Conceptual Framework of the SAWIT Tool

The SAWIT security awareness tool was built by applying Nonaka and Takeuchi's knowledge conversion SECI model, as it is described in [25].

The entire knowledge transformation process by using the SAWIT tool is shown in Figure 5.

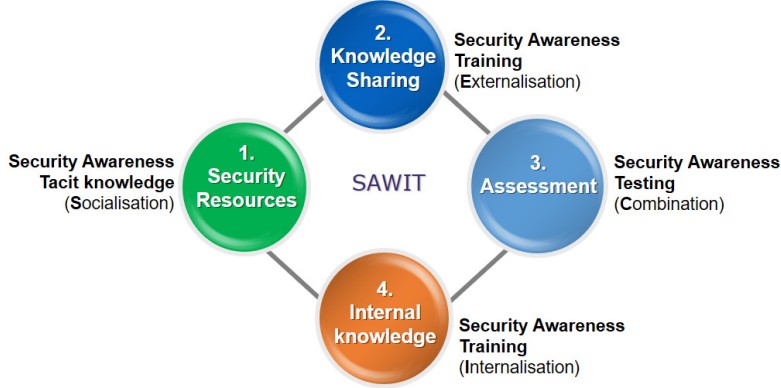

**Figure 5.** SAWIT conceptual model.

In order to gain an overall picture of their security level, the employees perform security pre-testing (*Socialization*) by using the SAWIT Security Resources tool (Figure 3). This stage is given by the organization and serves as a motivational factor for using the SAWIT tool. At that time, the tacit knowledge of employees can be measured and appropriate training (if needed) will be proposed. Security awareness training in areas where weaknesses have been identified (*Externalization*) is performed by using the SAWIT Knowledge Sharing tool (Figure 2). In this way, the security messages that are made explicit to the other employees become security awareness material. The third phase in the SECI knowledge transformation process is *Combination*, realized by using the SAWIT Assessment tool (Figure 4). Since explicit knowledge needs to be made tacit to the employees, after the material has been presented, the employees take a short test in order to measure to what extent the message has been internalized (*Internalization*). Finally, the actual behavior of employees is measured to examine whether their security behavior has changed because of their exposure to the SAWIT tool.

The security knowledge transformation of employees performed by the SAWIT tool employs the awareness raising method. It is aligned with cyber security awareness [13] and shows the movement through "perception", "comprehension" and "projection" in order to increase awareness levels ranging from (i) basic perception of important data, to (ii) interpretation and combination of data into knowledge to (iii) ability to predict future events and their implications. It is the basis for employees to understand the challenges associated with the secure use of information technology. It will help them to access their current knowledge, thus identifying any shortcomings and weaknesses, resulting in the acquisition of the required knowledge to be competent and confident using technology, as described in Korovessis et al. [17]. Having this in mind, the SAWIT tool aims to:

- Establish a structural approach so that an awareness program can add value to the organization/individual while contributing to the field of information security.
- Provide the means, so existing employees' knowledge is measured, providing insight into those areas where security knowledge is lacking through guidance on further knowledge creation.
- Provide an approach that employees can use in a structured and modular fashion so that security knowledge and skills can be built up over time.
- Include efficient presentation and interactivity methods so that employees are more engaged and eventually an appropriate level of knowledge retention is achieved.
- Provide a web-based system that employees will be able to access anywhere by using minimal information technology resources.

## 5. Evaluation

The main aim of the SAWIT tool is to improve security awareness among employees. Because of that, it is necessary to measure the improvement of employees' behavior regarding the detection and protection of security threats in cyber space. According to Parson et al., there is a strong, positive relationship between knowledge, attitude and behavior, or in other words, as the knowledge of safe information security behavior grows, this in turn impacts on better information security behavior [36]. Namely, the SAWIT tool should increase employees' knowledge, particularly about how to defend themselves against potential cyber-attacks in the workplace.

To gain initial insight into how users perceive and experience SAWIT, a formative evaluation was conducted. Jeske and Schaik state that university students are especially relevant for intervention, because they will soon start working, and if they do not have sufficient knowledge about cyber security, this knowledge gap should be addressed in training schemes within organizations [37]. This was the motivational trigger to engage voluntary students in finding out for themselves how useful the SAWIT tool is in improving security awareness and to evaluate SAWIT. Formative evaluation was carried out with 22 volunteer students from the University of Belgrade (20 undergraduates and 2 postgraduates). Although SAWIT is not completed yet, and its functionality is being improved, the participants experimented with the current version in order to learn how it might assist them in their future work.

The moderators demonstrated the use of the SAWIT tool to the participants. Having done so, the moderators then instructed the students to run the SAWIT tool, completing tasks which corresponded to the application scenarios described in Section 4. Finally, the participants completed questionnaires about SAWIT.

In creating our survey, we consulted the guidelines for creating questionnaires about evaluation of usefulness and ease of use software's system and user interfaces [38,39]. Beside this, we also adapted some questions from [17].

The questions in the questionnaire were related to the following criteria: relevance, organization of information, perceived usefulness and ease of use. The questions were designed on a 5-point Likert scale (the possible answers ranging from 1 to 5, from strongly disagree to strongly agree, or such like), but free-form qualitative judgments and comments were also encouraged.

According to the majority of the participants, SAWIT is quite relevant for cyber security, is also useful for security principles (86% participants Strongly Agree or Agree (SAA)), has appropriate topic coverage from the security perspective (77% SAA), provides in-depth analysis of security problems (73% SAA), is a good basis for security awareness (82% SAA) and is beneficial for team-work (91% SAA), as it is shown in Table 1.

**Table 1.** Likert scale questionnaire results.

| | Question | Agree/ Strongly Agree (%) | Neutral (%) | Disagree/Strongly Disagree (%) |
|---|---|---|---|---|
| Relevance | Benefits for team | 90.91 | 9.09 | 0.00 |
| | SAWIT is useful for security principles. | 86.36 | 9.09 | 4.55 |
| | SAWIT has appropriate topic converge from security perspective. | 77.27 | 18.18 | 4.55 |
| | SAWIT in-depth analyses security problems. | 72.73 | 13.64 | 13.64 |
| | SAWIT is good basis for security awareness. | 81.82 | 13.64 | 4.55 |
| Organization of information | Information is easily found. | 81.82 | 13.64 | 4.55 |
| | Information is structurally organized. | 81.82 | 13.64 | 4.55 |
| | Information is logically organized. | 81.82 | 13.64 | 4.55 |
| | Information is intuitively organized. | 81.82 | 13.64 | 4.55 |
| | Systems in the quiz were covered in the material. | 77.27 | 18.18 | 4.55 |
| Perceived usefulness | The material is clearly presented. | 81.82 | 13.64 | 4.55 |
| | The used vocabulary is adequate. | 77.27 | 18.18 | 4.55 |
| | Content makes my engagement relevant for learning. | 86.36 | 9.09 | 4.55 |
| | I found collaboration with colleges useful. | 90.91 | 4.55 | 4.55 |
| | I am satisfied with the system | 81.82 | 13.64 | 4.55 |
| Ease of use | SAWIT is easy to use. | 81.82 | 18.18 | 0.00 |
| | SAWIT is user friendly. | 81.82 | 13.64 | 4.55 |
| | I can easily recover after mistake. | 77.27 | 13.64 | 9.09 |
| | SAWIT can be used without written instructions. | 77.27 | 18.18 | 4.55 |
| | SAWIT is not inconsistent. | 81.82 | 13.64 | 4.55 |

The vast majority of the participants were very satisfied with the organization of information and stated that the information can be easily found (82% SAA), is structurally (82% SAA), logically (82% SAA) and intuitively organized (82% SAA) and post-assessment questions were covered in the material (77% SAA). On evaluating the usefulness of SAWIT, the participants said that the material is clearly presented (82% SAA), the vocabulary is adequate (77% SAA), the content is relevant for learning (86% SAA), collaboration with colleges is useful (91% SAA) and they are satisfied with the system (82% SAA). In analyzing ease of use, the participants said that SAWIT is easy to use (82% SAA), is user friendly (86% SAA) and one can easily recover after making a mistake (77% SAA). The participants also stated that SAWIT can be used without written instructions (77% SAA) and is not inconsistent (82% SAA).

As the SAWIT is a prototype model for security awareness, one participant suggested that in the final and complete version, more units should be added, with different levels of privileges for the user. One participant said that he liked the opportunity to add materials and suggested that it would be useful to add more real-world examples for social engineering in the form of videos, as well as video instructions for some units (more technical ones). One participant said that some specific vocabulary may be explained in more detail, not only by definitions but with real life examples. Two participants had problems using SAWIT, because they had a basic level of English language, and suggested that a future version of the toolkit should be in Serbian. One participant stated that they liked the ability to collaborate with colleagues and felt freer to ask colleagues rather than administrators, and this may also be of value to others in the group.

## 6. Related Work and Discussion

There are many materials available for security awareness in different forms, such as textbooks, infographics, video-lectures, games, and online courses (e.g., cybrary.it, cyberaces.org, edx.org), but most of them are focused on technical users. In addition, they are usually time-consuming, and very few have a user-centric approach to provide the end user with what he/she actually needs to know. The complexity and overwhelmingly technical aspects of information security awareness programs often present obstacles to self-learning for non-technical employees. On the other hand, although there is awareness training for end users, it is often limited to one specific threat such as ransomware [40], or phishing [41–43]. Besides that, most training programs fail to expand the learning content and offer no possibilities for collaboration or the ability for an ontological search.

Korovessis et al. presented the prototype for an information security toolkit for raising awareness [17]. Although their prototype has a similar concept to SAWIT, as they have both a pre-test and post-test to evaluate learning outcomes, there is no available collaboration between colleagues or a semantic search.

Sbityakov et al. proposed InCAT (Intelligence-based Cybersecurity Awareness Training), which connects the domain of cyber security awareness training with that of cyber threat intelligence [44]. The system is powered by the Watson Knowledge Studio platform and is still in the experimental phase. Although this is very promising, there is no collaboration between users, and the user is not able to add learning objects.

Work presented in [45] shows the system that collaboratively combines information from traditional and nontraditional sensors to produce new, relevant signatures. They also used the collaborative approach, but this approach is directed to detecting malicious traffic. On the other hand, the SAWIT collaborative approach is focused on education of employees in the area of cyber security in order to improve security awareness.

There is a good attempt to gather the knowledge from heterogeneous data sources [46]. This approach proposed the use of data fusion to enhance the defensive capabilities of the network and aid in the development of situational awareness for the security analyst. Unlike SAWIT, this approach does not support the collaborative learning since the analyst has only a limited (human) capacity to observe and interpret cyber security details.

Very interesting work is presented in [47] exploring the important conceptual issue of how to manage uncertainty and risk in cyber situational awareness. This approach described by two case studies is focused on individual system administrator, unlike the SAWIT tool, which proposes a collaborative learning approach.

To the best of our knowledge, other tools for security awareness training offer no possibilities for semantic annotations and collaborative learning, which is valuable in enhancing information retrieval and improving interoperability.

## 7. Conclusions

In the paper, the SAWIT toolkit was presented as the prototype of a web-based application for improving security awareness. Knowledge sharing among employees increases employees' awareness, and the SAWIT offers such an opportunity.

The SAWIT is hierarchically organized and has a structured approach. Firstly, we implemented the *human aspects of security* module, and in the future, we plan to implement more modules. With the expansion of security breaches and new threats, new objects/modules can be easily added to the SAWIT. Collaboration with colleges was shown to offer a more stimulating learning environment, and in that way, users do not become trapped in an exhausting boring environment. Moreover, providing real world examples in the SAWIT makes it more effective for employees, and they become more aware that security breaches may be real. The SAWIT may be used in continuous employee learning and development programs, which employees attend periodically and where they are continuously evaluated by measuring information security culture. To gain an initial impression of the SAWIT,

a group of students were introduced to it, and formative evaluation was carried out. The vast majority of the participants stated that the toolkit was easy to use and presented in an efficient and engaging way and that it was relevant for raising security awareness.

Although it is necessary for employees be knowledgeable about the subject in order to achieve secure behavior, it has also been reported that this was not enough to reduce insecure behavior [30,47–50]. Accordingly, our plan for future work would be to improve SAWIT by training and testing employees in real life situations. Aside from this, in the future, it will be valuable to analyze learning analytics so as to maximize training outcomes. This will provide better evidence of each user's learning style (e.g., when and how they prepare for tests, which lessons were difficult for them, who is the most active), which is very valuable information for improving learning content in the future. It was shown that users are more security-aware when they perceive more benefits from those behaviors and that they tend not to follow safe practices when costs increase [51]. We are also planning to develop a localization version of our software, because in some organizations, especially with older employees who are not familiar with the English language, the Serbian language (local) will be a necessity for its usage.

**Author Contributions:** Conceptualization and methodology A.K. and S.D.R.; software and validation S.D.R. and A.K.; investigation and resources A.K. and S.D.R.; writing S.D.R. and A.K.; supervision A.K. and S.D.R.; All authors have read and agreed to the published version of the manuscript.

**Funding:** This research received no external funding.

**Conflicts of Interest:** The authors declare no conflicts of interest.

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
