# Peer review of "SAWIT—Security Awareness Improvement Tool in the Workplace"

_applsci, doi:10.3390/app10093065_

Round 1

Reviewer 1 Report

Overall the paper presents an interesting solution for making users in a workplace more security aware. The paper is well presented and and generally portrays an understandable narrative but would benefit from some additions and amendments.

Section two and three discuss security awareness and related research, however there is no mention of seminal work by Micah Endlsey in Situational Awareness or its application to the Cyber field by authors such as Tadda, Ulrik Franke, Cyril Onwubiko

Example papers....

  • Endsley, M. R. (1995). Toward a Theory of Situation Awareness in Dynamic Systems. Human Factors, 37(1), 32–64.

  • Tadda GP, Salerno JS. Overview of cyber situation awareness. In: Cyber Situational Awareness. Springer; 2010. pp. 15e35.
  • Tamassia

  • Franke, U., & Brynielsson, J., 2014. Cyber situational awareness – a

systematic review of the literature. Computers & Security, 46, 18–31.

doi:10.1016/j.cose.2014.06.008

The tool evaluation in section six at present is limited. The tool was evaluated using a questionnaire however a more rigorous evaluation using a recognised usability tool such as SUS (Software Usability Score) would provide a quantifiable usability score for the tool. Comparison with another similar tool (perhaps identified in section three) would also strengthen this section. In addition, evaluating the tool using metrics such as accuracy or efficiency (See ISO 9241-11) would provide more rigour

Author Response

  • We improve section 2 with model of situational awareness, and their applications.
  • In the section of evaluation, we provide more information about our guidelines and papers for creating questionnaire. Although, suggestion is good for comparing our results with similar tool, it  was not possible (different questions) and one paper have not analysed evaluation of software. 
  • Also we improved related work section

Reviewer 2 Report

The topic is interesting but the paper is very uninspiring. For example, the reference sources are limited and a reporting style is evident and does not really bring out the essence of what is being proposed. It is necessary I think for the author(s) to look more deeply at the subject matter and to at times offer value judgements. It can be argued that the SAWIT tool is the work of the author(s) but it is not clear how it was devised and how it will be utilized. Referring back to the literature  review, the subtopics need more clarity and depth so that the tool proposed is underpinned and placed in context. The cybers security issues need to be looked at in more depth and so does the learning organization approach because it is not possible to assume that employees will use the SAWIT tool if they are not influenced to do so. Security awareness is key and there is a good literature on the topic and more can be written about  cybers security from both a technological and human issues perspective. The SECI model is relevant but more attention needs to be given to helping the reader make the link with it and convince  them it will help as regards the use of the SAWIT model and its implementation. There does need to be a stronger link between knowledge utilization and section 4, The SAWIT Tool Experience. The latter is very basic and reflects  the fact that the paper is written like a report and not an intellectually challenging article. Section 5, The Conceptual Framework of the SAWIT Tool also needs to be strengthened. The positioning of the paper needs to be defined better and also, it would benefit from real world examples that can demonstrate its usefulness or possibly external staff can be interviewed to talk through some of the complexities. The paper is timely but overall, it is rather underdeveloped.p

Author Response

In Section 4, line 196-199 it is explained how SAWIT is connected with security  awareness

Organisation push  employees under stress to use SAWIT, and stress may be main trigger for using cyber security awareness (section 4, 237-241)

Organisation may be motivational factor for using SAWIT tool (section 4, line 285-287)

Connection between cyber security awarness and SECI (section 4, line 298-301)

Reviewer 3 Report

I  this paper, a security awareness tools "SAWIT" is proposed. 

However, almost half of this paper seemed to be the citation of previous works, so I can not feel novelty. More detailed description of "SAWIT" is required.

And, In the abstract, it announced "it is an innovative artificial intelligence framework...", however I can not find the matter of artificial intelligence in this paper.

As for the system, this tool will be able to achieve collaborative learning, however it is not mentioned how to prepare for the initial contents.

Author Response

  • Line 216-220 it is explained how SAWIT uses artificial inteligence
  • Link between SAWIT and artificial intelligence is mored detail explained in section 4, line 216-219. 
  • Organisation gave initial content for SAWIT 

Round 2

Reviewer 2 Report

The paper reads better. Some parts will require minor editing. It seems the word etc in the text should be replaced by "for example".

Possibly need to cite the author(s) s opposed to just have a source number.

More attention could have been given to the uniqueness of the work and how it will improve cyber security overall. This can be followed up in another paper perhaps.

Author Response

The paper reads better. Some parts will require minor editing. It seems the word etc in the text should be replaced by "for example".

  • Line 48 etc is replaced (or be the victim of the phishing attack)
  • Line 352 etc is deleted

Possibly need to cite the author(s) s opposed to just have a source number 

In conclusion we added opposite opinions, about unsuccessful security awareness attempts (line 398-401):

"Although employees being knowledgeable on the subject is necessary to achieve secure behavior, it has also been reported that was not enough in reducing insecure behavior [47,48,49,50,51]. Accordingly, our plan for future work would be to improve SAWIT by training and testing employees in real life situations"

More attention could have been given to the uniqueness of the work and how it will improve cyber security overall. This can be followed up in another paper perhaps.

On the end of section of Related work and discussion, line 377-379

“To the best of our knowledge, other tools for security awareness training offer no possibilities for semantic annotations and collaborative learning, which is valuable in enhancing information retrieval and improving interoperability.”
